# Occupational Exposure to Biological Agents in a Typical Restaurant Setting: Is a Photocatalytic Air Purifier Helpful?

**Matteo Ratti *** , **Daniele Ceriotti** , **Rabia Bibi** , **Andrea Conti** and **Massimiliano Panella**

Department of Translational Medicine (DiMeT), Università del Piemonte Orientale, 28100 Novara, Italy; 10036607@studenti.uniupo.it (D.C.); 20036124@studenti.uniupo.it (R.B.); andrea.conti@uniupo.it (A.C.); massimiliano.panella@uniupo.it (M.P.)
* Correspondence: matteo.ratti@uniupo.it

**Abstract:** According to many national legislations, biological agents represent an occupational hazard that must be managed in order to ensure safety at workplace. Bioaerosols have been associated to many pathological conditions but, despite many efforts, precise threshold limit values (TLV) are still undefined. We planned and conducted an environmental study concerning a typical restaurant that aimed to evaluate: (1) the occupational exposure to bacterial and fungal bioaerosol; (2) the efficacy of a photocatalytic air purifier device in mitigating such exposure. This observational study evaluated two dining rooms (Area 1 and Area 2) of a restaurant which can be considered typical during two consecutive weeks. Based on a national protocol, we monitored total bacterial and mycotic loads searching for two typologies of bacteria, psychrophilic bacteria (environmental contamination) along with mesophilic bacteria (human or animal origin source), and two types of fungi, mold and yeast. Baseline total bacterial load was 346.8 CFU/m$^3$ for Area 1 and 412.9 CFU/m$^3$ for Area 2. When the sanitizing device was operative, the total bacterial load decreased to 202.7 CFU/m$^3$ (−41.50%—*p* value: <0.01) for Area 1 and to 342.2 CFU/m$^3$ (−17.10%—*p* value: 0.06) for Area 2. Considering the fungal load, the mean baseline value was 189.7 CFU/m$^3$ for Area 1 and 141.1 CFU/m$^3$ for Area 2. When the device was kept on, the total fungal load was 108.0 CFU/m$^3$ (−43.10%—*p* value: 0.055) for Area 1 and 205.0 CFU/m$^3$ (+45.30%—*p* value: 0.268) for Area 2. Our findings supported the conclusion that, concerning the occupational risk derived from biological agents, a typical restaurant should be considered relatively safe. In order to mitigate or limit any possible increase of such risk, a photocatalytic device may be helpful, but not against the pollution caused by mold or yeasts. Our research also reaffirmed the need of further research assessing the kind of relationship between diseases and exposure levels, before considering the need of setting precise threshold limit values.

**Keywords:** occupational risk; bioaerosol; photocatalytic device; indoor air quality





## 1. Introduction

It is widely recognized that bioaerosols represent an occupational hazard, with an established association to diseases like asthma, hypersensitivity pneumonitis, and the sick building syndrome [1–4]. In fact, many occupational studies have demonstrated associations between bacterial endotoxin exposure and health effects including both reversible and permanent obstruction (asthma or Chronic Obstructive Pulmonary Disease), other respiratory symptoms or skin or systemic allergies. These were observed in a large variety of occupational environments (e.g., farms, animal industries, food processing factories, waste and compost industry), which differ each other mainly by exposure levels [5–7].

The major constituents of bioaerosol are microorganisms like bacteria, viruses or fungi, along with other organic matter such as skin dust, animal excreta, and pollen grains. Following a specific European Union directive [8], many national workplace safety institutes of developed countries currently provide local laws and regulations about the evaluation of risks derived from biological agents. In fact, even though to date any attempt to identify precise threshold limit values (TLV) have been unsuccessful, some recommendations

have been provided: for instance, a European 1993 document reported thresholds for non-industrial indoor environment categories based on a review of environmental studies [9]. However, the range among different categories is very wide (from <50 to >2000 Colony Forming Units/m$^3$ or CFU/m$^3$), and the authors warned that this categorization was not reflecting a health risk, since no direct relation between microbial load and pathological conditions  had not been demonstrated at the time. Notably, the threshold separating a low category from an intermediate one was set at 500 CFU/m$^3$ both for bacteria and fungi.

Nevertheless, the legislation about work safety in some countries report TLVs about indoor pollution from microorganisms that far exceed this value: for example, for the total bacterial load 5000, 10,000, and 4500 CFU/m$^3$ values were proposed in eastern European countries, Germany, and Finland, respectively [10–12]. Similarly, for the total fungal load 5000, 10,000, and 750 CFU/m$^3$ have been recommended in eastern European countries, Germany, and Brazil, respectively [10,11,13]. In Italy, the most recent document from the National Institute for Insurance against Accidents at Work (INAIL) recommends the limits provided by the European Collaborative Action [14] but this document is dated 2010 and is still to be updated. The rationale behind all these proposed values is contrasting a possible occupational risk, so that when a measurement exceeds the limit, the employer is obliged by law to implement corrective measures in order to ensure safety at the workplace.

As the human presence and activities are demonstrated to be major sources of indoor bioaerosol, the facilities like school or restaurants, in which individuals commonly stay for long, are expected to suffer more from indoor pollution [15–17]. For instance, a recent study concerning a school reported elevated bacterial (81 times) and fungal load (15 times) when occupied with respect to vacant condition. Moreover, by comparing indoor emissions to ventilation (infiltration of outdoor air), the authors found that indoor emissions are the dominant source of bacteria and fungi, accounting for circa 97% of indoor air bacteria, and  91% of indoor air fungi, and concluded that indoor emission sources must be controlled to significantly impact human bacterial and fungal exposure in the indoor air [18].

In order to limit exposure to biological agents, several strategies can be used:  for instance, negative pressure systems have been successfully used in a Central Sterile Supply Department, where the CFU/m$^3$ of bacterial charge dropped from 273 to 116 [19]. Another study evaluated the use of different personal protective equipment (PPE) in a wastewater treatment plant. The authors found a Gram negative load of 114.60 $\pm$ 63.02 CFU/m$^3$ and concluded that "...although the bioaerosol concentrations were generally regarded as safe according to existing standards, these bioaerosols' health risks were still unacceptable", and reaffirmed the need of such PPEs [20].  Also air cleaning systems, high-efficiency particulate air (HEPA) filters and air purifier systems such as the photocatalytic (PCO) ones have been demonstrated effective in containing bioaerosol pollution in various working settings [21–25]. Notably, the PCO technology has been recently tested successfully against SARS-CoV-2 virus [26,27].

Since in recent years the occupational safety is becoming more and more a public health concern (as demonstrated by the updating of national legislations about the topic and by the SARS-CoV-2 pandemic) we planned and performed an environmental study concerning the indoor air pollution from bioaerosol in a local restaurant. Based on a sectoral study from the National Internal Revenue Agency [28], the characteristics of the facility are reputed to be representative, as well as the urban setting in which it is located. The aim was to describe the occupational exposure represented by such bioaerosol and to assess the efficacy of a photocatalytic air purifier in decreasing the microbial load, and, ultimately, improve safety at the workplace.

## 2. Materials and Methods

### 2.1. Study Design and Setting

This observational study was performed in a real-life setting: a local restaurant situated in the city of Novara, Piedmont, Northern Italy. Two sections of the facility were designated as Area 1 (dining internal room, 45 m$^2$) and Area 2 (entrance hall plus dining area, 47.25 m$^2$).

The heating, ventilation and air conditioning system (HVAC) was built in 2018 when the restaurant opened, equipped with HEPA filters 99.95%. Annual maintenance procedure were regularly performed, as prescribed by law. According to the national regulation UNI 10339/1995 [29] and the declaration of conformity of the company that built the system, the outddor supply air flow rate is 10 L/s per person for the restaurant dining rooms. Figure 1 shows the planimetry of the restaurant with the defined Areas and the HVAC system. The facility had 45 seatings. Based on the information derived from the 2012 technical note of the restaurant sectoral study conducted by the National Internal Revenue Agency, the majority of restaurant facilities in Italy have a small comparable mean surface (91 square meters) and an average of 50 seatings [28].

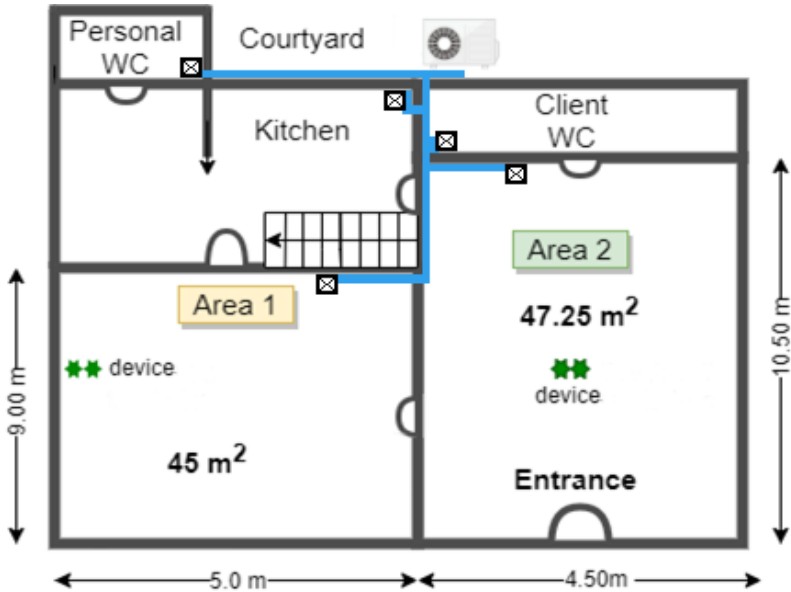

**Figure 1.** Planimetry of the restaurant facility. The location of the sanitizing devices is marked with two green stars. The HVAC system has been highlighted in blue.

### 2.2. Timeline of the Study

The study lasted for two weeks, from 10 January 2022 to 23 January 2022, for a total of 13 days. During the first week, two PCO air purifying devices were installed in Area 2 only. Then, the following week, the Areas were switched, with Area 1 having the working devices while Area 2 remained unsanitized.

### 2.3. Microbiological Monitoring

The basic microbiological parameters were evaluated according to the protocol provided by INAIL (Italian National Institute for Insurance against Accidents at Work) [14]. Based on this protocol, we monitored total bacterial and mycotic loads searching for two typologies of bacteria: psychrophilic bacteria, which grow at around 22 °C (range 15–30 °C) and are currently considered an indicator of environmental contamination [30], along with mesophilic bacteria, that have optimal growth around 37 °C (range 25–40 °C) and are considered an indicator of human or animal origin source [31]. We also searched for mycotic loads including molds and yeast: they are mesophilic organisms that can grow from 0 to 50 °C . These are common environmental indicators of high humidity, reduced ventilation, and poor air quality, and also opportunistic pathogens [32–34].

### 2.4. Sanitizing Device

OYA-N (Inspira s.r.l., Novara, Italy) is a photocatalytic air purifier programmed for active sanitizing of the indoor environment. It works by photocatalitic oxidation of air pollutants with the emission of ozone  and its working mode is adjustable by the user depending on the size of the surface to be treated. Photocatalysis is an established

technology against indoor air pollution, both for volatile organic compounds (VOC) and microorganisms [35,36]. The photocatalytic cell produces continuous oxidation of water molecules in the atmosphere leading to the neutralization of the VOCs smaller than 0.002 m. In addition, the UV (Ultraviolet) light that activates the cell has a direct germicidal effect. The device is equipped with a fan of 23 cm of diameter, with its speed adjustable (4 levels). Following the producer's specifications, the speed was set at level 2, which corresponds to circa 250 $m^3$/h or 2 vol/h. The controlled emission of ozone intensifies the oxidation process accelerating the sanitization of the environmental air. According to the producer, the device is effective against bacteria, viruses, fungi, and its safety has been validated by the achieved CE certification that is reported in Supplementary Materials File S1). With regard to the ozone, the machine has a release rate of less than 0.1 parts per million (ppm), therefore not exceeding both European and US occupational limits values [37,38]

### 2.5. Sample Collection and Incubation

The samples were collected by an external company (B. And Partners Safety Environment Consultancy s.r.l., Magnago, Milan, Italy) specialized in the field of environmental analysis and safety, and certified by the Italian Ministry of Health. The sample campaign was conducted according to the mentioned INAIL protocol [14]. The technicians employed an active SAS (Surface Air System) single-stage sampler with orthogonal impact (TRIO.BAS™ Trio, Orum International s.r.l., Milan, Italy), that consisted of three aspirating chambers and relative slots for agar Petri plates. The SAS technology has been detailed and analyzed in depth by Whyte et al. [39]. In brief, the air containing the microbes is aspirated and accelerate through a hole and direct towards a nutrient agar surface of a plate. As the air turns away from the agar surface, the microbe-carrying particles that cannot follow the flow are impacted. The procedure was carried out according to the ISO 14698-1 "Cleanrooms and associated controlled environments" and the ISO 14698-2 "Evaluation and interpretation of bio-contamination data" documents. The TRIO.BAS devices product sheet, along with the application notes reporting the validated usage Standard Operating Procedures, are reported in the Supplementary Materials File S2. The TRIO.BAS sampler have been employed recently in similar studies [40].

The samples were collected twice a day, one after the lunch service between 14:00–15:00 and one after dinner between 22:00–23:00 for 13 consecutive days, excluding "Day 7" (weekly resting day). The sampler was put at the center of the room at 1.5 m of height as shown in Supplementary Materials Figures S1 and S2. The SAS air sampler typology was proved to yield comparable results with respect to other instruments such as Bourdillon slit sampler [41] or Andersen two-stage sampler [42]. The sampling time was 15 min at a flow rate of 100 L/min.

The 90 mm Agar or Saboraud culture mediums contaminated by the sample were then sent to the company laboratory and incubated in a thermostat incubator for an appropriate period: the plates for psychrophilic bacteria were incubated at $(22 \pm 1)$ °C for 48 h, while the ones for mesophilic bacteria were incubated at $(36 \pm 1)$ °C for 72 h. Molds and yeast were incubated at $(25 \pm 1)$ °C for 5–7 days. The sum of the loads (both mesophilic and psychrophilic) resulted in the total bacterial load value. Similarly, the sum of mold and yeast charges was represented as the total mycotic load. Consistently with the literature, the microbial contamination was expressed as Colony Forming Units (CFU) per $m^3$. Moreover, a positive hole correction for 400 holes has been applied to the results following a consolidated methodology, to avoid underestimation due to the fact that more than one particle impacted the medium through the same hole [43].

### 2.6. Statistical Analysis

The statistical analysis was performed with R version 4.1.0 (R Core Team, Vienna, Austria) [44] and R Studio version 2021.09.2 (RStudio, PBC, Boston, MA, USA) with tidyverse package [45,46]. The total bacterial load and mycotic load along with their subsets were described with mean, standard deviation (SD), and relative percentage change. The

measurements were paired according to the same day and hour of the week, thus accounting for the fact that the restaurant, like the majority of the others, was generally more crowded during weekends and evenings. Furthermore, the restaurant opens to the public only in two daily timeframes (from 12 to 14 and to 19 to 23), while in the other periods the facility is solely accessible to the staff. We adopted this approach to mitigate the effect of crowding to our results, also considering the limited study timeframe of two weeks. The mean of the differences between the sanitized and non-sanitized values by Areas were then tested by a paired two-tail *t*-test. We considered statistically significant *p* values less than 0.05. The study protocol was preliminary approved by the local ethical committee of the "Ospedale Maggiore della Carità di Novara" with the assigned code EC 232/21 on 7 October 2021.

### 3. Results

During the two weeks, we collected and processed a total of 46 samples. Area 1 had a mean of $10.7 \pm 2.1$ occupants per service, while Area 2 totaled $10.2 \pm 1.9$ individuals. Table 1 summarizes the values of both the bacterial and fungal samples analysed.

**Table 1.** Means and standard deviations of microbial load sampled in the two Areas of the restaurant. The values in bold were considered statistically significant.

| Area/Charge | Non-Sanitized Measure—Mean (SD) | Sanitized Measure—Mean (SD) | Mean of the Differences in Paired Values (95%CI) | Relative Percentage Change Compared to Unsanitized Measure | *p* Value |
|---|---|---|---|---|---|
| Area 1 – bacterial | 346.8 (104.6) | 202.7 (70.7) | 154.5 (63.2–245.9) | −41.50% | **<0.01** |
| *of which mesophilic* | *183.5 (60.8)* | *107.3 (44.3)* | | | |
| *of which psychrophilic* | *163.3 (50.7)* | *95.5 (35.9)* | | | |
| Area 1—fungal | 189.7 (94.3) | 108.0 (86.6) | 92 (−2.28–186.3) | −43.10% | 0.055 |
| *of which yeast* | *32.7 (15.2)* | *21.5 (11.7)* | | | |
| *of which mold* | *157 (89.9)* | *86.5 (76.2)* | | | |
| Area 2—bacterial | 412.9 (150.9) | 342.2 (94.5) | 70.2 (−4.67–145.0) | −17.10% | 0.063 |
| *of which mesophilic* | *233.1 (102.6)* | *186.7 (50.9)* | | | |
| *of which psychrophilic* | *179.8 (72.0)* | *155.5 (54.0)* | | | |
| Area 2—fungal | 141.1 (127.6) | 205.0 (117.9) | −70.2 (−203.65–63.2) | 45.30% | 0.268 |
| *of which yeast* | *30.4 (19.9)* | *32.8 (13.3)* | | | |
| *of which mold* | *110.7 (114.3)* | *172.2 (114.7)* | | | |

The mean baseline total bacterial load was 346.8 CFU/m$^3$ for Area 1 and higher (412.9 CFU/m$^3$) for Area 2. The variability was higher in Area 2 when compared to Area 1 (150.9 vs. 104.6 CFU/m$^3$). None of the measurements of Area 1 surpassed the category level of intermediate, while in 3 occasions Area 2 registered a value of over 500 CFU/m$^3$. When the sanitizing device was operative, the total bacterial load was 202.7 CFU/m$^3$ for Area 1 and 342.2 CFU/m$^3$) for Area 2. The variability was still higher in Area 2 in respect to Area 1 (94.5 vs. 70.7 CFU/m$^3$), even though set at a lower level. Only one measurement surpassed the intermediate threshold, in Area 2. The relative reduction of the total bacterial charge was found to be −41.50% (*p* value: <0.01—statistically significant) for Area 1 and −17.10% (*p* value: 0.06—not statistically significant) for Area 2. Figure 2 shows the boxplot of the total bacterial load values retrieved. The boxplots and line charts of the psychrophilic and mesophilic load are included in the Supplementary Materials (Figures S3, S4, S7 and S9).

Considering the fungal load, the mean baseline value was 189.7 CFU/m$^3$ for Area 1 and lower (141.1 CFU/m$^3$) for Area 2. The standard deviation was slightly higher in Area 2 when compared to Area 1 (127.6 vs. 94.3 CFU/m$^3$). When the device was kept on, the total fungal load was 108.0 CFU/m$^3$ for Area 1 and 205.0 CFU/m$^3$) for Area 2. The variability was higher in Area 2 in respect to Area 1 (117.9 vs. 86.6 CFU/m$^3$), but with

with values comparable to the non-sanitized measurements. Only two samples surpassed the intermediate threshold, both collected in Area 2. However, they were considered outliers since they fell outstide the 1.5 times of interquartile range boundaries. Nevertheless, the statistical plan did not allow the removal of outliers, in order to be sure of the effect of the device. The relative reduction of the total fungal charge was found to be −43.10% (*p* value: 0.055—not statistically significant) for Area 1 and +45.30% (*p* value: 0.268—not statistically significant) for Area 2. Figure 3 shows the boxplot of the total fungal load values collected. The boxplots of the mold and yeast load are included in the Supplementary Materials (Figures S5, S6, S8 and S10).

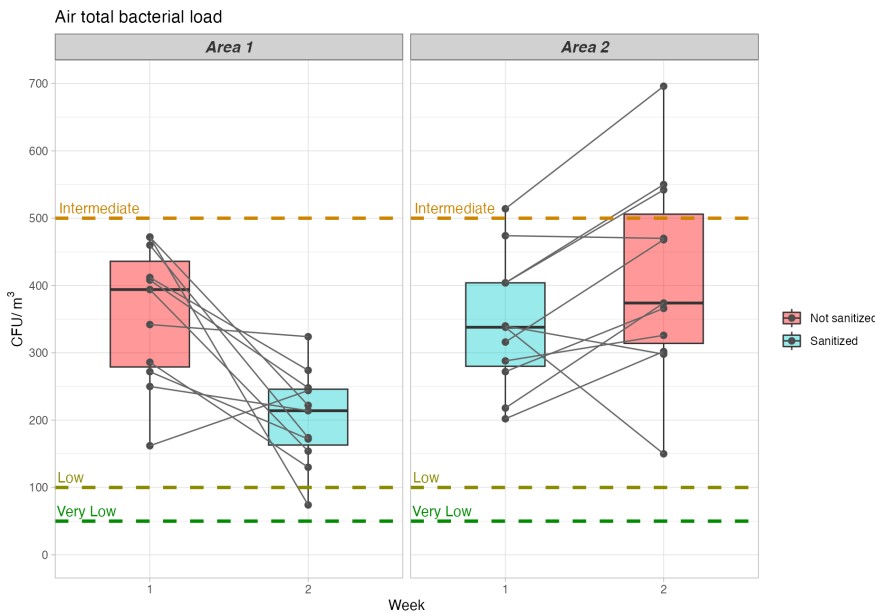

**Figure 2.** Boxplot of the total bacterial load retrieved in the restaurant. The measurements were paired according to the day of the week and the hour of the sample. The reference categories are taken from [9,14].

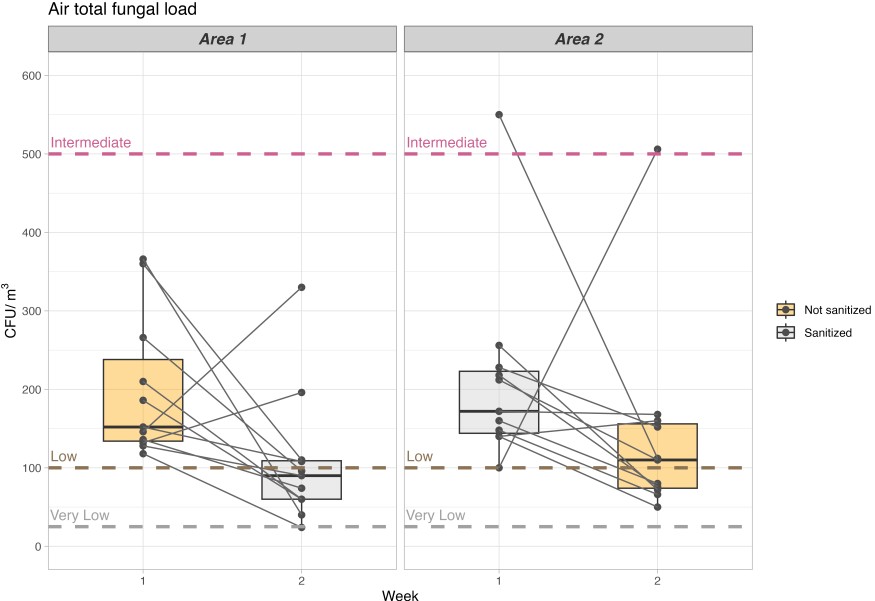

**Figure 3.** Boxplot of the total fungal load retrieved in the restaurant. The measurements were paired according to the day of the week and the hour of the sample.The reference categories are taken from [9,14].

## 4. Discussion

Our primary aim was to describe the risks derived from bioaerosol in a typical restaurant in urban setting and, secondarily, to assess if the use of a PCO air purifier could be of any help in mitigating such risks. Considering the exposure, even though no precise threshold value limit have been established by the relative scientific research, the legislation of many developed countries currently forces the employer to evaluate the occupational risks and adopt proper countermeasures in case of specific exposure to biological agents. For this reason, we characterized the biological pattern of the restaurant during two normal operational weeks and found that almost all the measurements were below the intermediate category set in the European collaborative action document [9] that still acts as a reference in the field (Table 2). In fact, these values are also reported in the protocol of the Italian National Institute for Insurance against Accidents at Work (INAIL) about microbiological sampling at the workplace [14].

**Table 2.** Categories of indoor microbiological charge as defined by European collaborative action Report n°12—Biological Particles in indoor environments [9].

| Category | Houses (Bacteria— CFU/m$^3$) | Non Industrial Environment (Bacteria— CFU/m$^3$) | Houses (Fungi— CFU/m$^3$) | Non Industrial Environment (Fungi— CFU/m$^3$) |
|---|---|---|---|---|
| Very Low | <100 | <50 | <50 | <25 |
| Low | <500 | <100 | <500 | <100 |
| Intermediate | <2500 | <500 | <1,000 | <500 |
| High | <10,000 | <2000 | <10,000 | <2000 |
| Very High | >10,000 | >2000 | >10,000 | >2000 |

Note: these values do not reflect a health risk, unless a direct relationship between exposure (measured by CFU/m$^3$) and any disease will be proved.

Regarding the bacterial load, we noticed that Area 1 had a lower baseline average value with respect to Area 2 (346.8 CFU/m$^3$ vs. 412.9 CFU/m$^3$). Considering that the mean occupants were roughly the same (the number of individuals are in fact directly correlated with the bioaerosol levels [15,16]), we think that this finding may be due to the different characteristics of the rooms: in fact, the entrance hall belongs to Area 2, and, since the outdoor environment is known to represent a major source of indoor pollution [47–49], the repeated opening and closure of the door along with the more frequent comings and goings of the individuals probably contaminated the air in this Area in a relative higher proportion.

A recent study evaluated the condition of a commercial diary plant, finding a median value of 265 CFU/m$^3$ (range: 40–980) for bacteria and 165 CFU/m$^3$ (40–390) for fungi. The authors concluded that no occupational risk exists at this level of contamination [50]. Another study concerning a waste sorting facility reported an average concentration of total bacteria of 4347 CFU/m$^3$ (SD = 2439), considered excessive and resulting in a recommendation of an extensive use of Personal Protective Equipment [51]. A third interesting recent evaulation of the working condition in wastewater treatments plants in Denmark reported that 14% and 34% of the personal exposure were exceeded for endotoxin (>50 EU/m$^3$) and bacteria (>500 CFU/m$^3$). The authors also defined an hazard index using a 500 CFU/m$^3$ threshold for bacteria contamination: the geometric mean found in the facility of 299 CFU/m$^3$ was considered low occupational risk [52]. Therefore, considering the literature about the topic, it seems that the limit of 500 CFU/m$^3$ currently act as a threshold for a low occupational risk and that, based on our results, we can reasonably conclude that a typical restaurant does not pose critical risks for the workers and no adjunctive PPE may be considered. In other words, the restaurant is a relatively safe workplace (at least in the dining rooms). However, it is worth mentioning that only a few studies used bacterial and fungal concentrations to describe the health effects of bioaerosols. According to a systematic review, the bioaerosols exposure-response relationship has not been clearly

defined yet due to lack of studies with valid dose-response data, diversity of measuring methods for microorganisms and bioaerosol-emitting facilities, heterogeneity of health effects or insufficient exposure assessment [53]. The same authors also retrieved several indicator parameters and exposure concentrations among the analyzed studies, for different typologies of facilities. We agree with them that health-related exposure limits are urgently needed (especially in approval procedures).

Regarding the PCO device efficacy, we noticed favorable results only in Area 1 and relative to bacterial contamination, with a relative reduction of 41.50%. This reduction, even if not much relevant in terms of safety, is way less compared to the efficacy measures found in laboratory setting: recent studies about PCO devices found bacterial load decreases ranging from 55% to 76.5% [21,54,55]. We reasonably think that this discrepancy is normal when shifting from ideal conditions to real world experiments. Even though in Area 2 the reduction did not reach statistical significance, we noticed that the variability (SD) of the measures decreased from 150.9 to 94.5. From an occupational risk evaluation point of view, this result could be reputed satisfactory, because it limits the times that an individual could be at risk: in fact, only one sample exceeded the threshold of 500 CFU/$m^3$ when the device was functional.

Although the PCO technology is proved to be effective against fungi [56], we noticed no effect towards them in our study: a possible explanation could reside in the intrinsic higher resistance of this kind of microorganisms with respect to oxidants. This finding is supported by a study which conclusion was that fungal cells are intrinsically resistant to the penetration of free radicals because of pigments and thick walls [57]. Moreover, a study by Edward et.al demonstrated that it took much more time to kill *A. Niger* spores (72 h for 90%) compared to *E. coli* bacteria (one hour for 99.9%) [58]. This could mean that for a typical restaurant we may need more than two devices at work or to increase the operational time to obtain efficacy against fungi. The analysis of the fungal load showed some values that were considered outliers. Nevertheless, we did not exclude them from the analysis because the predefined statistical plan did not allow to do so. As a matter of fact, excluding observations in such a small sample may increase type I error to an unacceptable level. This cautionary rule may have resulted in an underestimation of the effect of the device towards fungi.

*Limitations*

The risk derived from bioaerosols is intrinsically linked to the typology of bacteria/fungi. In fact, the exposure we analysed is only a part of the overall risk assessment procedure, since it is composed of both exposure to a hazard and the relative damage it may cause. Even though we distinguished between psychrophilic and mesophilic bacteria (and between mold and yeast) for detecting the probable origin of the contamination, no qualitative assessment of the bacteria/fungi was performed. This means that even though we are confident in the 500 CFU/$m^3$ limit, some residual risk could exists that is derived from the qualitative presence (rather than quantitative) of specific pathogens like, for instance, *L. pheumophila* or *Aspergillus* spp. We must reaffirm that the categories set in the EU document [9] are representative of indoor environment samples taken thirty years ago and are not associated to a health risk unless a direct relationship between exposure levels and any disease will be proved. Even though these values are commonly used as a reference in the field of environmental indoor air quality studies since many years, our conclusions might be invalidated by future studies. Finally, since our sample was small, we recommend caution in generalizing our results until more data will be generated by similar future studies. Nevertheless, we believe our findings may be of help during the design phase of further real world studies or in case of summarising secondary studies.

## 5. Conclusions

The results of our study supported the conclusion that, concerning the occupational risk derived from biological agents, a typical restaurant should be considered relatively

safe because the levels of exposure to bioaerosols in our experiment remained low. In order to mitigate or limit any possible increase of the relative risk, a photocatalytic device may be helpful, but not against the pollution caused by mold or yeasts bioaerosol. Our research also reaffirmed the need of further research assessing the kind of relationship between diseases and exposure levels, before considering the need of setting precise threshold limit values.

**Supplementary Materials:** The following supporting information can be downloaded at: https://www.mdpi.com/article/10.3390/safety9040081/s1, Figures S1–S10; Tables S1–S3; Files S1–S3.

**Author Contributions:** Conceptualization, M.P.; investigation, D.C.; formal analysis, M.R.; writing—original draft preparation, M.R.; writing—review and editing, R.B.; data curation, R.B.; validation and supervision, A.C. All authors have read and agreed to the published version of the manuscript.

**Funding:** This research received no external funding.

**Institutional Review Board Statement:** The ethical committee of "Azienda Ospedaliera Maggiore della Carità" (Novara, Italy) approved the study with protocol number 1010/CE (Study label n. CE 232/21) on 7 October 2021.

**Informed Consent Statement:** Informed consent was obtained from all subjects involved in the study.

**Data Availability Statement:** The full dataset of the obtained measures is included in Supplementary Materials Table S1.

**Conflicts of Interest:** The authors declare no conflict of interest.

## Abbreviations

The following abbreviations are used in this manuscript:

| | |
|---|---|
| TLV | Threshold Limit Values |
| COPD | Chronic Obstructive Pulmonary Disease |
| CFU | Colony Forming Units |
| INAIL | Istituto Nazionale Assicurazione Infortuni sul Lavoro |
| PPE | Protective Personal Equipment |
| HVAC | Heating, Ventilation, and Air Conditioning |
| HEPA | High Efficiency Particulate Air |
| PCO | Photocatalytic Oxidation |
| VOC | Volatile Organic Compounds |

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
