# Peer review of "Occupational Exposure to Biological Agents in a Typical Restaurant Setting: Is a Photocatalytic Air Purifier Helpful?"

_safety_

Round 1
Reviewer 1 Report
Comments and Suggestions for Authors
The study analyzed bacteria and fungi in two areas of an Italian restaurant using a protocol from the Italian National Institute for Insurance against Accidents at Work. They specifically looked at airborne culturable bacteria that thrive in cold temperatures and bacteria that prefer normal temperatures and also at culturable fungi. Then they evaluated culturable bioaerosol loads before and after the operation of OYA-N (Inspira s.r.l., Novara, Italy), which is a photocatalytic air purifier. The samples were collected twice daily, once after lunch and once after dinner service times, for two weeks. Bioaerosols were found to be relatively low in the study, and the air purifier helped to reduce them somewhat, which is statistically significant. The major weakness in the study design was the very low sample size. The study only considered one small restaurant of approx. 100 square meter size and collected data for only two weeks. The researchers did not identify the microorganisms, making the health significance of the data uncertain. The authors may consider the following comments to improve their manuscript.
1. Title: The use of the word ‘risk’ in the title is ambiguous because the authors did not conduct occupational personal exposure assessment and health risk assessment following standard protocols. The authors may consider changing the word.
2. Introduction: The study rationale is not adequately described. Why did the authors consider it important to address the risks of workers' health and safety in restaurant workplaces due to exposure to psychrophillic and mesophillic bacteria and fungi? Please review the research literature on this topic and justify your study's hypothesis and objectives.
3. P. 3, line 106: Ozone release from these air purifiers could be a health concern for the workers and customers. Please provide the ozone release rate data.
4. P. 4, line 115: Please provide more information about the active single-stage sampler with orthogonal impact (TRIO.BAS™ Duo). Did anyone use this sampler in other similar research studies? Please cite those references. Also mention the sampling time and numbers of samples collected per run. Is there any positive hole conversion associated with this sampler similar to commonly used Andersen type samplers? Please provide this information.
5. Discussion section: Do you think 40% decrease of bioaerosol load has strong health significance? Please compare your data with similar studies conducted in the labs.
6. Figures 2 and 3, Y axis – correct the legend spellings – CFU and superscript of 3 in m3.
Author Response
Dear Reviewer, please refer to the attached document.
We thank you for the time spent in evaluating our work.
Kind Regards,

Reviewer 2 Report
Comments and Suggestions for Authors
Becausue the distribution of the measured values are not normal and there are some extreme value, the authors should consideration to perform non-parametric tests (eg. Welconson signed rank test) or take log transformation to avoid the outliers affected the results.
Author Response
Dear Reviewer,
please refer to the attached document.
We kindly thank you for the time spent in evaulating our work.
Kind Regards,

Reviewer 3 Report
Comments and Suggestions for Authors
The study is well-performed and the manuscript well-elaborated. I have no major nor minor remarks.
Author Response

(The authors gave the same response as above.)

Reviewer 4 Report
Comments and Suggestions for Authors
General comments: This was a well-conducted study of two areas of a single restaurant. Please exercise caution when generalizing to other restaurant environments beyond this study. Please supply more justification for applying the paired t-test. I am not convinced the data should be paired. There are guidelines for indoor air quality, for instance, the American Society of Heating, Refrigerating and Air-Conditioning Engineers (ASHRAE) recommends 3.8 L/s-person outdoor are supply rate for restaurant dining rooms in ASHRAE standard 62.1-2022. The publication is available on line: https://ashrae.iwrapper.com/ASHRAE_PREVIEW_ONLY_STANDARDS/STD_62.1_2022
Better characterizing the studied restaurant would aid readers in determining similarity to an indoor space they are considering.
Specific comments:
|
Line |
Comment |
|
33 |
Please change “…environments…” to “…environment…” |
|
35 |
Please define the CFU/m3 unit when you first introduce it. |
|
66 |
Change “…characteristic…” to “…characteristics…” |
|
70 |
Change “…at workplace.” To “…at the workplace.” |
|
Figure 1 |
Was there any heating, ventilating and air conditioning (HVAC) system? Any forced air, or filtration? |
|
98 |
Please describe more about the air cleaner. Did it use a fan to pull air through the device? What was the estimated air changes per hour (flow rate of the device divided by the room volume)? |
|
113 |
Please capitalize, “Italian”. |
|
118 |
Please change “…13 consecutively days,” to “…13 consecutive days,” |
|
135 |
Why did you use a paired t-test? You explained that you matched by day. Please provide more justification. Were those days similar in terms of expected microbial load for some reason? Because of weather, sunlight, temperature, relative humidity, customer traffic, or why? If you have that data that might affect microbial load, then you should model based on those variables. I don’t know that you have justified using paired t-test. |
|
149 |
Please change “tough” to “though”. |
|
Figure 2 |
Again, I don’t think the pairing of data is justified unless you have some other evidence. |
|
162 |
How did you justify treating high readings as outliers? Did you perform a q-test, or have other notes? |
|
164 |
I think you mean fungal rather than bacterial. |
|
Figure 3 |
I think this figure helps show that the fungal data are not really paired. Also, there are outliers, but that is similar to other biological aerosol sampling reports. This supports one of the reasons against setting a limit for bioaerosols. There is large variability, even in carefully executed sampling campaigns. |
|
215 |
You may also add other variables which might affect the biological load in a restaurant, such as air flow, HVAC, filtration, sources of biological aerosols, humidity, etc. |
|
219 |
Please change “…intrinsecally…” to “…intrinsically…” |
|
232 |
Please reword your conclusions. I am not convinced that your study of two restaurants shows that this occupational environment is safe. You could qualify the statement with better description of the two test rooms in the one tested restaurant. Was there heat or air conditioning? HVAC? Last changing of air filter, if present? What is the estimate of air flow? Lastly, you argue for a precise TLV. Your study upholds a widely held position that a precise TLV is misleading and difficult to apply. |
|
247 and following |
Please look at your abbreviations section. You applied the capitalization randomly. Please make it uniform. For instance, “threshold limit values” is all lower case, but “Colony Forming units” has a mixture of upper and lower case. “Volatile Organic Compounds” is all upper case, but repeated. |
Comments on the Quality of English Language
Your use of the English language was very good. I did provide a few comments to assist you.
Author Response

(The authors gave the same response as above.)

Round 2
Reviewer 1 Report
Comments and Suggestions for Authors
I have reviewed the revised manuscript and the responses from the authors. The revised manuscript is much improved and responses are adequate.
Author Response
Dear Reviewer, we thank you for the time spent in evaluating our work
We added the Institutional Review Board approval details.
Kind Regards,
Reviewer 2 Report
Comments and Suggestions for Authors
The authors have responded my comments appropriately.
Author Response

(The authors gave the same response as above.)

Reviewer 4 Report
Comments and Suggestions for Authors
Thank you for considering my comments. Your paper looks very good.
Author Response

(The authors gave the same response as above.)
